# Backpropagation with Continuation Callbacks: Foundations for Efficient and Expressive Differentiable Programming

**Fei Wang**
Purdue University
West Lafayette, IN 47906
wang603@purdue.edu

**James Decker**
Purdue University
West Lafayette, IN 47906
decker31@purdue.edu

**Xilun Wu**
Purdue University
West Lafayette, IN 47906
wu636@purdue.edu

**Grégory Essertel**
Purdue University
West Lafayette, IN, 47906
gesserte@purdue.edu

**Tiark Rompf**
Purdue University
West Lafayette, IN, 47906
tiark@purdue.edu

## Abstract

Training of deep learning models depends on gradient descent and end-to-end differentiation. Under the slogan of *differentiable programming*, there is an increasing demand for efficient automatic gradient computation for emerging network architectures that incorporate dynamic control flow, especially in NLP.

In this paper we propose an implementation of backpropagation using functions with callbacks, where the forward pass is executed as a sequence of function calls, and the backward pass as a corresponding sequence of function returns. A key realization is that this technique of chaining callbacks is well known in the programming languages community as *continuation-passing style (CPS)*. Any program can be converted to this form using standard techniques, and hence, any program can be mechanically converted to compute gradients.

Our approach achieves the same flexibility as other reverse-mode automatic differentiation (AD) techniques, but it can be implemented without any auxiliary data structures besides the function call stack, and it can easily be combined with graph construction and native code generation techniques through forms of *multi-stage programming*, leading to a highly efficient implementation that combines the performance benefits of define-then-run software frameworks such as TensorFlow with the expressiveness of define-by-run frameworks such as PyTorch.

## 1   Introduction

*Differentiable programming* (Olah, 2015; LeCun, 2018) refers to a programming model where neural networks are truly functional blocks with data-dependent branches and recursion, while at the same time being trainable with backpropagation and gradient descent (Rumelhart et al., 1986). A programming model of such generality requires both expressivity and efficiency from the backpropagation framework. However, the current generation of tools such as TensorFlow (Abadi et al., 2015), and PyTorch (Paszke et al., 2017) trade off one for the other.

Inspired by the pattern of forward and backward passes, this paper proposes an implementation of backpropagation using functions with callbacks. Each elementary operation becomes a function call. The forward computation for this operation is performed on function entry, and the backward computation on function exit. In between, the result of the forward computation is passed to a

callback, which executes the downstream (forward and backward) computations (Figure 1). The use of callbacks provides modularity and enables programmers to chain arbitrary operations together. While programming in this style with explicit callbacks is of course cumbersome, a key realization is that this programming pattern has been well known in the programming languages community for more than 50 years under the name *continuation-passing style* (CPS) (van Wijngaarden, 1966), and there is a simple and well-studied transformation that converts any program into CPS (Fischer, 1972).

This approach achieves the same flexibility as other *define-by-run* reverse-mode automatic differentiation (AD) techniques (Wengert, 1964; Speelpenning, 1980) and naturally extends to loops, subroutines, and recursive functions. Unlike other approaches, however, it can be implemented without any auxiliary data structures (often called *trace* or *tape*). We implicitly use the call stack as our data structure, with the benefit that the memory is automatically managed and out-of-scope data are freed when no longer needed. Using *delimited continuations* and shift/reset control operators (Danvy and Filinski, 1990), we can make the callbacks implicit, too, and provide an implementation of reverse-mode AD solely through operator overloading.

Our approach can further be combined with existing graph construction and native code generation techniques to provide an expressive *define-then-run* computation model, including in-graph functions and recursion. In particular, we employ an orthogonal concept called *multi-stage programming* (staging, Taha and Sheard (2000)). Inspired by the natural observation that most programs operate in separate stages due to data availability and frequency of operation (Jørring and Scherlis, 1986), programming language researches developed tools where a program can be partially evaluated, with code generated for the unevaluated part. The generated code can be in a different (potentially low-level) language, thus removing abstractions (objects, higher-order functions) and improving efficiency (Taha and Sheard, 2000). Specifically, by utilizing Lightweight Modular Staging (LMS) (Rompf and Odersky, 2010), we create a highly efficient and expressive framework dubbed Lantern which supports both unrestricted control flow as found in PyTorch, as well as the computation graph reification in, e.g., TensorFlow.

We explain the requisite programming languages concepts and present evaluation results as follows:

- Section 2 shows how delimited continuations naturally support reverse-mode AD.
- Section 3 explains how multi-stage programming orthogonally brings efficiency.
- Section 4 evaluates Lantern and demonstrates efficiency and expressivity of our framework.

Finally, Section 5 discusses related work and offers concluding thoughts.

## 2 Differentiable Programming and Reverse-Mode AD

### 2.1 Reverse-Mode AD, Explained

Let $v_1, v_2, ..., v_k$ be the nodes in a computation graph $\mathcal{G}$ in a topological ordering (i.e., every node corresponds to some function $f_i$ that depends on results of earlier, parental nodes as parameters). For neural networks, $v_k$ reflects the loss $\mathcal{L}$, which is the target of optimization. During reverse-mode AD, the forward pass first traverses the nodes from $v_1$ to $v_k$, computing the result (value) of each node. The backward pass then traverses the nodes from $v_k$ to $v_1$, computing the gradient $d\mathcal{L}/dv_i$ for each node, which defines the effect of tiny changes of $v_i$ on the value of $\mathcal{L}$. While $d\mathcal{L}/dv_k$ is 1.0, $d\mathcal{L}/dv_i$ for $i < k$ are calculated by the chain rule:

$$\frac{d\mathcal{L}}{dv_i} = \sum_{j \in \text{Out}(i)} \left( \frac{\partial f_j}{\partial v_i} \right)^{\text{T}} \frac{d\mathcal{L}}{dv_j}$$

Here, $\text{Out}(i)$ defines the output nodes of node $v_i$ in graph $\mathcal{G}$, and $\partial f_j / \partial v_i$ is the Jacobian matrix of the partial derivative of $f_j$ to $v_i$.

### 2.2 Reverse-Mode AD as Functions with Callbacks

Normally, reverse-mode AD is implemented with the help of auxiliary data structures. For instance, the small example

$$v_1 = 0.5 \quad v_2 = 0.4 \quad v_3 = v_1 + v_2 \quad v_4 = v_2 * v_3 \quad v_5 = \tanh(v_4)$$

can be represented as the computation graph in Figure 1 (top).

The gray arrows (above each node) form the forward pass, and the red arrows (below each node) form the backward pass. Each node is represented as a rounded square, with the upper half containing the formula for computation (N/A for nodes with initial values), and the lower half containing the value (left) and the gradient (right). The formulas for the backward pass are labeled on the red arrows, and gradients from multiple arrows are summed together. Operations of formulas can be computed by a graph iterator which performs a forward pass first, then a backward pass. This is similar to the implementation of TensorFlow and PyTorch, though TensorFlow creates new nodes for backpropagation by explicit graph transformation/optimization.

Inspired by the "There and Back Again" (Danvy and Goldberg, 2005) pattern of reverse-mode AD, a key observation is that we can profitably perform these operations as a sequence of function calls, one for each elementary operation (Figure 1, bottom). In the lower section of the figure, the executor of every operation is explicitly labeled. The first $v_1 + v_2$ operation is performed by the caller of the whole function (possibly grad, denoted $g$). $g$ calls the first callback $k_1$, which handles the $v_2 * v_3$ operation, and calls the second callback $k_2$. $k_2$ then computes $\tanh(v_4)$ and calls the last callback $k_3$. $k_3$ only needs to set the gradient of $v_5$ as 1.0. After $k_3$ returns, $k_2$ updates the gradient of $v_4$ by the chain rule, and returns to $k_1$, which updates the gradients of $v_2$ and $v_3$. Upon $k_1$'s return, $g$ updates the gradients of $v_1$ and $v_2$. The scopes of each function/callback are also highlighted by dashed boxes, showing nested scopes of the chain of callbacks. Note that although nodes are retained in the figure for callbacks, it is easy to see that the values and gradients can be saved on the function call stack: no auxiliary heap-allocated data structures are needed.

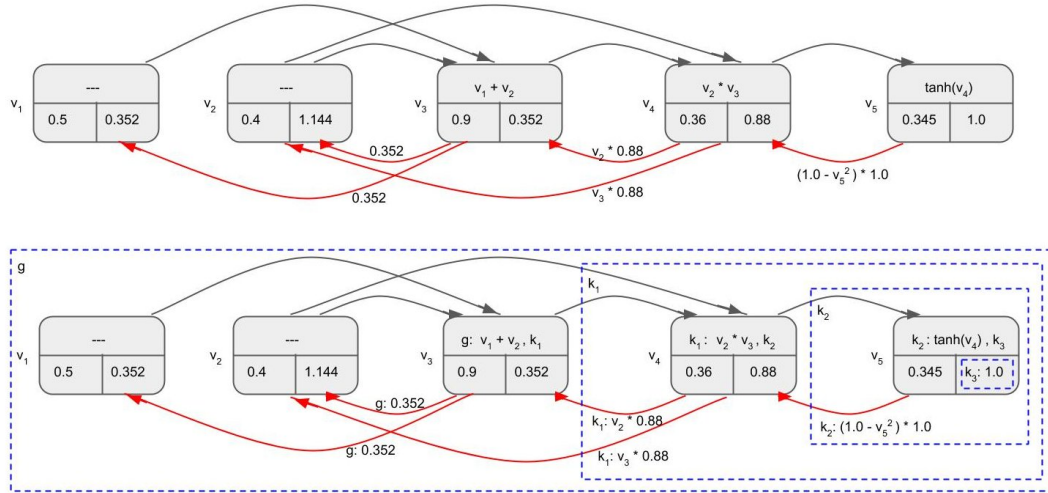

Figure 1: Reverse-Mode AD represented as graph nodes (top) and reverse-Mode AD via callbacks (bottom)

With this dual view of chains of nodes and nested function calls (Figure 1), we can see that the call path implements the forward propagation, and the return path implements the backward propagation. Inspired by this idea, we show the Scala implementation of this callback version of reverse-mode AD in Figure 2.

## 2.3 Implementation Using Operator Overloading

Our first implementation in Scala is mechanical, directly following the drawing in Figure 1. As shown in the left column of Figure 2, we define a class NumR with two fields: an immutable value x, and a mutable gradient d. Each operator in NumR takes a callback k which consumes the intermediate NumR (y) as a parameter and handles the following forward pass and the leading backward pass. Once the callback k returns, the gradient of y (the correct value of y.d) should have been computed. Then the operator updates the gradients of the dependent values as side effects, using the value of y.d. On the right column is the definition of the grad operator, an example, and the expected unit test. Note that in the definition of grad we provide the final callback of (r => r.d = 1.0), which is to set up the gradient of the final NumR as 1.0. To aid in presentation, the occurrences of callbacks appear shaded.

```scala
// differentiable number type
class NumR(val x: Double, var d: Double) {
  def +(that: NumR) = { (k:   NumR=>Unit) =>
    val y = new NumR(x + that.x, 0.0);  k(y)
    this.d += y.d; that.d += y.d
  }
  def *(that: NumR) = { (k:   NumR=>Unit) =>
    val y = new NumR(x * that.x, 0.0);  k(y)
    this.d += that.x * y.d; that.d += this.x * y.d
  }
  ...
}
```

```scala
// differentiation operator
def grad(f:NumR => (NumR=>Unit)=>Unit )(x:Double)={
  val z = new NumR(x, 0.0)
  f(z)(r => r.d = 1.0)
  z.d
}
// example: 2*x + x*x*x
val df = grad { x =>
  (2*x) (y1=> ( x*x )(y2=> (y2 *x )(y3=> y1 + y2)))
}
// unit test
forAll { x => df(x) = 2 + 3*x*x }
```

Figure 2: Automatic Differentiation in Scala: reverse-mode AD by callbacks and operator overloading (left), and the grad function definition and use case (right). Handling of continuations is highlighted. Code first appeared in Wang and Rompf (2018)

Unfortunately, the example (last shaded box in Figure 2) is coded in a rather cumbersome way, simply because we must explicitly construct the callbacks for each step (implicit conversion of Int to NumR is elided). A natural question, then, is: Could this be simplified or automated?

### 2.4 Implementing Reverse-Mode AD with Continuations

This idea of introducing callbacks for *every function result* is actually a well-known program transformation, named *continuation-passing style* (CPS), which has been studied in the PL community for more than 50 years (van Wijngaarden, 1966).

The concept of continuations is ubiquitous in programming: an if-branch is the choice between two continuations, an exception or goto is an abortion/change of continuation, etc. However, in a normal, "direct style" of programming, continuations are maintained implicitly by function calls/returns and other control flow. By contrast, CPS manages control flow by passing continuations explicitly (every function has an extra parameter called continuation $k$). For instance, while a direct-style function returns its result directly to the calling function, a CPS function takes as an argument "the rest of the computation" as a function (i.e, continuation), and calls the continuation with the result as a parameter. CPS is often used in compilers as an intermediate representation of programs (Appel, 1992). The transformation of direct-style programs into CPS is also a well-known procedure Fischer (1993), shown below (Figure 3, upper).

Transformation to continuation-passing style:

$$
\begin{array}{rcl}
[\![\text{if } (e_1) \ e_2 \text{ else } e_3]\!] \ k &=& [\![e_1]\!](v_1 \Rightarrow \text{if } (v_1) \ [\![e_2]\!] \ k \text{ else } [\![e_3]\!] \ k) \\
[\![\text{while } (e_1) \ e_2; \ e_3]\!] \ k &=& \text{def loop}() = \{[\![e_1]\!] \ (v \Rightarrow \ \text{if } (v)[\![e_2]\!] \ \text{loop else } [\![e_3]\!] \ k)\}; \text{loop}() \\
[\![\text{def } f(n_1, ...) = e_1; \ e]\!] \ k &=& \text{def } f(n_1, ..., k') = \{[\![e_1]\!] \ k'\}; \ [\![e]\!] \ k \\
[\![e(e_1, ...)]\!] \ k &=& [\![e]\!] \ (v \Rightarrow ([\![e_1]\!] \ (v_1 \Rightarrow (... \Rightarrow v(v_1, ..., k)...))))
\end{array}
$$

Transformation of delimited control operators shift/reset:

$$
\begin{array}{rcl}
[\![\text{shift}(k \Rightarrow e)]\!] \ k' &=& \text{def } k(r, k'') = k''(k'(r)); [\![e]\!](x \Rightarrow x) \\
[\![\text{reset}(e)]\!] \ k' &=& k'([\![e]\!](x \Rightarrow x))
\end{array}
$$

Figure 3: Program Transformation between direct style (left) and CPS (right). $[\![e]\!] \ k$ denotes a program $e$ in direct style, transformed with given continuation $k$.

The rules in Figure 3 transform direct-style programs to CPS, where the continuations are always maintained as tail calls, which never return to the callers. However, this is insufficient for the callbacks needed in reverse-mode AD, as these callbacks must return. This can be achieved through the use of *delimited continuations* (Felleisen, 1988), which, as the name suggests, are continuations up to certain boundaries, defined by the control delimiters. When arriving at the boundaries, the continuations return to *their* caller, possibly with return values. In that sense, delimited continuations are more like normal functions, and they do not have to be tail calls. The remaining key difference is that delimited continuations are constructed from part of the program.

Delimited continuations can be generated from direct-style programs supplemented with *control operators*. Several forms of control operators exist, but we will use the pair with the closest relation to CPS, named `shift`/`reset` (Danvy and Filinski, 1990). The formal rules are shown in Figure 3 (bottom), which lay out the form of transformations for the `shift` and `reset` operators. Modern tools (Rompf et al., 2009) further simplify the transformations for delimited continuations to selective CPS transformation, where program fragments without `shift`/`reset` are kept in direct style (no need for the $k''$ parameter in the $k$ function in `shift` transformation rule).

Generally speaking, the `reset` operator defines the boundary of the delimited continuations, while the `shift` operator captures the delimited continuations. Their roles can be further explained using the following toy example.

```
val a = 1 + reset { 10 +  shift { k => k(k(100)) + 1000 }  }
```

The delimited continuation is the program between `shift` and `reset` (the shaded box above), which can be embodied by replacing the `shift` construct (the white box above) as function parameter, and rewriting the `reset` block as the function, i.e., continuation (the shaded box below), and then passed to the `shift` block.

```
val a = 1 + { (k => k(k(100)) + 1000  (x => 10 + x)  }
```

Then the delimited continuation is captured by the `shift` construct, as the continuation parameter k in the `shift` block. The final result is then the evaluation of the body of `shift`.

```
val a = 1 + { (10 + (10 + 100)) + 1000 } = 1121
```

In this way, the program is still written in direct style (with `shift` and `reset` operators). However, the automated transformation will reorganize it into CPS format, realizing delimited continuations. Thus, the cumbersome example of Figure 2 can be simplified by using `shift` and `reset` in construction. We provide this implementation below (Figure 4).

```scala
// differentiable number type
class NumR(val x: Double, var d: Double) {
  def +(that: NumR) =  shift {(k:NumR=>Unit)=>
    val y = new NumR(x + that.x, 0.0); k(y)
    this.d += y.d; that.d += y.d
  }
  def *(that: NumR) =  shift {(k:NumR=>Unit)=>
    val y = new NumR(x * that.x, 0.0); k(y)
    this.d += that.x * y.d; that.d += this.x * y.d
  }
  ...
}

// differentiation operator
def grad(f: NumR => NumR @cps[Unit] )(x: Double) = {
  val z = new NumR(x, 0.0)
  reset { f(z).d = 1.0 }
  z.d
}
// example
val df = grad(x => 2*x + x*x*x)
// unit test
forAll { x =>
  df(x) = 2 + 3*x*x
}
```

Figure 4: Automatic Differentiation in Scala: reverse-mode using delimited continuations with `shift`/`reset` operators (left), and `grad` function definition and use case (right). Code first appeared in Wang and Rompf (2018)

In this figure, the occurrences of `shift`/`reset` and delimited continuations are again shaded. The `shift`/`reset` program transformation is handled by the Scala compiler accordingly (Rompf et al., 2009). The implementation of `NumR` with `shift`/`reset` operators is almost identical to `NumR` in Figure 2 (modulo added `shift`). Note that a `shift` operator returns a CPS-annotated type `A@cps[B, C]`, meaning that the continuation k in `shift` is of type (A ⇒ B), and the body of `shift` is of type C. When type B equals C, we denote it as `A@cps[B]`. Importantly, handling of continuations is confined to implementation logic and does not leak into user code (see the example in Figure 4).

Our approach has some similarity with the seminal paper by Pearlmutter and Siskind (2008) who also formalized reverse-mode AD in a functional style. However, several important aspects are substantially different. For one, their implementation uses nonlocal code transformations to return a pair consisting of a value and a backpropagator: $x \mapsto (v, dv/dy \mapsto dx/dy)$ for back propagation. We apply delimited continuations using `shift`/`reset` operators, which hide the nonlocal transformations from the developer, so that reverse-mode AD can be implemented purely via operator overloading. Their approach is purely functional (no variables are mutated during computations), which needs special care (a channel) if a lambda uses variables from an outer scope. On the other hand, we allow

limited mutation of gradients (gradient accumulation), which offers elegant implementation at the slight (and worthwhile, in our opinion) trade-off of functional purity. Moreover, all closures and mutable variables in our approach can be allocated on the stack, which serves as an implicit data structure for intermediate values. Other current approaches require at least some use of heap memory.

Higher-order gradients can also be computed with our approach. One technical caveat is that second-order shift/reset operators are not available in Scala, thus we cannot naively nest our gradient computations, though it can be achieved in a different language which supports higher-order shift/reset. However, even in Scala, we can get second-order gradients (Hessians) by combining reverse-mode AD with forward-mode AD. We elide forward-mode AD in this paper, as it can be easily implemented by operator overloading in many languages. By applying forward-mode AD on top of reverse-mode AD (changing the Double in the code snippets to a pair of Doubles, representing the value and tangent, respectively), we can efficiently compute Hessians (or the Hessian vector dot product, for any given vector).

## 3   Code Generation via Multi-Stage Programming

Via delimited continuations, we get an expressive and *define-by-run* framework, similar to PyTorch. However, TensorFlow and other *define-then-run* frameworks benefit from separating graph construction and graph execution into two stages, so that graph transformations/optimizations can be performed to target hardware-specific code (i.e., GPUs or TPUs). As such, we examine the possibility of utilizing this concept.

A key insight in understanding how to adopt this paradigm is that TensorFlow graph construction is similar to a 30-year-old PL concept called multi-stage programming (staging, Taha and Sheard (2000)). A TensorFlow program can blend normal Python code with graph construction, just like the well-established staging tool called Lightweight Modular Staging (LMS) (Rompf and Odersky, 2010) can blend normal Scala program execution with IR construction (this IR (intermediate representation) is not executed, but rather used to generate code for the next stage).

```
# graph construction          // graph construction        // generated code
import tensorflow as tf        import lms._                 float x0 = 0.0;
a = tf.constant(0)            val a: Rep[Float] = 0.0      while (x0 < 10) {
b = lambda i: tf.less(i, 10)  while (a < 10)                 x0 += 1
c = lambda i: tf.add(i, 1)      a += 1                     }
r = tf.while_loop(b, c, [i])  val r = a                    float x1 = x0;
```

Figure 5: TensorFlow graph construction (left), LMS IR construction (middle), and code generated from LMS (right).

We show a simple TensorFlow graph construction example and corresponding LMS code generation in Figure 5. Instead of tf.constant, LMS uses higher-order types (Rep[T]) to label IR constructions. All Rep-typed values (and computations depending on Rep-typed values) are treated as IR and translated into generated code, while all other typed values are treated as normal Scala expressions and are "staged away" from the generated code. Relying on type inference and advanced operator overloading, LMS also extends to built-in control flow constructs like if, for, and while, so that normal syntax with subroutines and recursion can be used, in striking contrast to the clunky TensorFlow API. In fact, the Rep types in LMS code are the only giveaways that any IR construction is taking place. We elide the mechanisms of code generation in LMS, as they are not a contribution of this paper but covered in a substantial body of relevant publications (Rompf and Odersky, 2010; Rompf, 2012; Rompf et al., 2012; Rompf and Odersky, 2012; Kossakowski et al., 2012; Ackermann et al., 2012; Ofenbeck et al., 2013; Rompf et al., 2013, 2015; Rompf and Amin, 2015; Rompf, 2016a,b; Ofenbeck et al., 2017; Amin and Rompf, 2018; Stojanov et al., 2018; Tahboub et al., 2018; Essertel et al., 2018).

Although we showcase LMS as the tool of staging, and shift/reset in Scala, it should be noted that these two concepts are supported in other languages as well: our design is not confined to Scala. For instance, shift/reset are common fare in certain dynamic languages in the Lisp/Scheme/Racket tradition, often implemented via stack-copying at runtime (Clinger et al., 1999). It would be very much feasible to implement shift/reset in Python; the "Stackless Python"[1] dialect already provides similar facilities. Efforts like AutoGraph (Moldovan et al., 2018) provide LMS-like staging mechanisms for a subset of Python.

There are also several choices in how to combine delimited continuations and staging. A program can be CPS transformed first, then staged to low-level languages. Otherwise, we can choose to stage the program to a medium-level language first (e.g., Scheme), do CPS transformation, and then compile it to low-level code (C/CUDA). Various degrees of engineering may be needed depending on the choice of languages and options, but no fundamental challenges should exist.

We choose to implement CPS-then-staging in Scala, merely out of convenience. With the requisite implementations in place, we have established an expressive framework capable of supporting branches, loops, and recursion, similar to the define-by-run style of PyTorch. However, our approach is actually define-then-run, which maintains a larger surface for analysis and optimization, like TensorFlow (but with in-graph functions and recursion). Aside from high-level optimizations among tensor operations that can be added in staging, our approach may benefit from general compiler optimizations as well, since the program after CPS transformation is no different from normal programs that are free of AD logic.

## 4    Evaluation and Case Studies

In this section, we validate our design by implementing and evaluating our prototypic framework, dubbed Lantern[2]. Lantern builds on the code in earlier sections, but supports handling tensor objects (multi-dimension arrays with common linear algebra operations such as element-wise operations with broadcasting, matrix multiplication, and convolution). The basic classes are shown below, with `Tensor` relating to `Double`, and `TensorR` relating to `NumR` in earlier code snippets. Note that for each `Tensor`, the data is `Rep` typed (as IR), but the shape is not (as it is known at staging time). Each `TensorR` object contains a value `x` and a gradient `d`, and operations on `TensorR` are implemented with `shift` operators providing access to delimited continuations.

```
class Tensor(val data: Rep[Array[Double]], val shape: Array[Int]) {...}
class TensorR(val x: Tensor, val d: Tensor) {...}
```

While some operations are linked to the OpenBLAS implementation, most operations are implemented as simple C++ loops. Even with such a naive backend implementation, Lantern demonstrates potential for being both expressive and efficient, at least for some small/medium-sized models running on a single CPU, as shown by comparing with PyTorch, TensorFlow, and DyNet (Neubig et al., 2017). To be complete, we plan to integrate with standard tensor compiler pipelines (e.g., XLA (TensorFlow team, 2018; Distributed (Deep) Machine Learning Community, 2018)) or with purpose-built compiler frameworks that directly extend LMS (e.g., Delite and OptiML (Sujeeth et al., 2014, 2011)) as future work.

### 4.1    Evaluation of Four Common Deep Learning Architectures

We selected four representative machine learning architectures for our evaluations: a vanilla Recurrent Neural Network (RNN), Long Short-Term Memory (LSTM), TreeLSTM, and a Convolutional Neural Network (CNN). Sample implementations of these in TensorFlow or PyTorch are readily available online, with either artificial or practical benchmarks. As a new deep learning framework that provides reverse-mode AD with a tensor API, our evaluation focuses on expressivity and efficiency, rather than model generalization.[3]

As shown in Figure 6, we compared Lantern with TensorFlow and PyTorch (DyNet implementation was only introduced for TreeLSTM for the benefit of autobatching). The training loss (not shown) in all architectures had similar decay, indicating that Lantern correctly implements backward propagation. We elected to only gauge the runtime of training loops, as that is the majority of computation. For vanilla RNN and LSTM, we evaluated at batch size 20. The training time for Lantern in both cases is less compared with that of PyTorch, and comparable to that of TensorFlow. For CNN, the evaluation was done at batch size 100, and Lantern performed similarly with PyTorch and TensorFlow (compiled from source with Intel® Math Kernel Library for Deep Neural Networks (Intel® MKL-DNN) support).

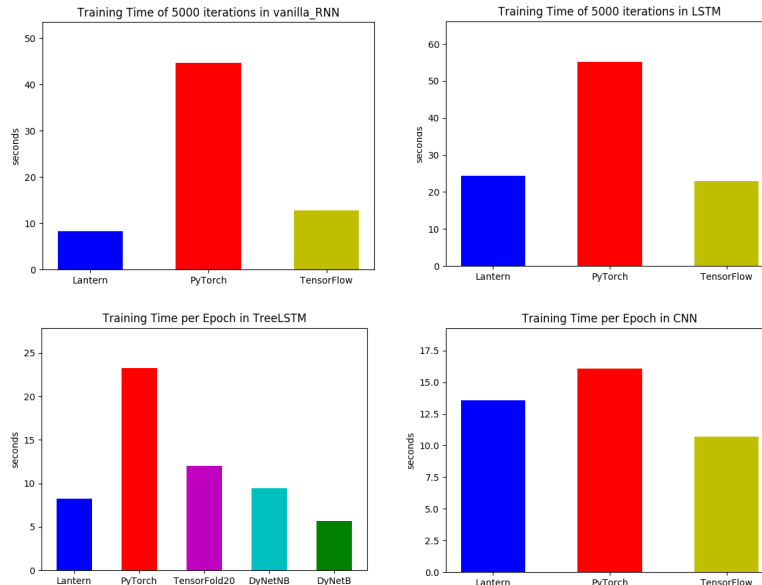

Figure 6: Comparison of training times for vanilla RNN (top left), LSTM (top right), TreeLSTM (bottom left), and CNN (bottom right).

We would like to give extra attention to the evaluation of TreeLSTM, which is adapted from Sentiment Classification using the dataset from the Stanford Sentiment Treebank (Chuang, 2013) following the work of Tai et al. (2015). Briefly, the model evaluates tree-structured parsed sentences (movie reviews) for sentiment (range 1 to 5).

$$h_i = \text{TreeLSTM}(\text{Embedding}(word), h_{i.\text{left}}, h_{i.\text{right}})$$

Here, $h_i$ is the hidden vector and the cell state (default when describing LSTM) associated with node $i$, and the Embedding is a large lookup table which maps each word to a 300-sized array, reflecting the semantic distances between all words in the vocabulary. TreeLSTM differs from a simple LSTM by taking two previous states, from both the left and right children. For leaf nodes, the previous states are zero, as is the embedding for non-leaf nodes. The hidden vector from each node can be used to compute a softmax of labels, thus generating a cross-entropy loss by comparing with the true label for each node. By training, the total loss (or average loss per sentence) should be reduced; thus the TreeLSTM learns to evaluate reviews in a parse-tree format.

```
// definition of loss function
def lossFun(root: Rep[Tree]) = {
  val init = (init_loss, init_hidden, init_cell)
  def f = FUN { node: Rep[Tree] =>
    if (node.isEmpty) init else {
      val (left, right) = (f(node.left), f(node.right))
      LSTM_core(left, right)              // return (new_loss, new_hidden, new_cell)
    }
  }
  val (outLoss, _, _) = f(root)
  outLoss                                 // only return the loss
}
// gradient update loop
for (n <- (0 until maxIter): Rep[Range]) {
  grad(lossFun(next_training_data()))     // gradients are updated as side effects
  gradient_descent()
}
```

Figure 7: TreeLSTM implementation in Lantern. FUN emits a differentiable recursive function.

This model is worth examination due to the fact that TreeLSTM is a recursive model (the computation graph is recursively and dynamically defined by the structure of the training data). In PyTorch[4] and

Lantern, this model can be easily expressed as recursive functions (see Lantern implementation in Figure 7), but Lantern's implementation is more efficient (Figure 6). However, such dynamic models are very hard to batch: one cannot simply add another dimension for batching, since each training datum may require a different computation graph. As such, both Lantern and PyTorch were run at batch size 1. Both TensorFlow and DyNet have partial solutions to this challenge.

TensorFlow cannot handle a recursive model easily, so the implementation[5] used TensorFlow Fold (Looks et al., 2017), a TensorFlow extension that statically modifies the computation graph based on the training data. Such a tool is more clunky and ad-hoc to use, but the benefit is that it allows for effective batching, since a unified static computation graph is constructed based on the training data. We evaluated TensorFold at batch size 20, and it indeed runs faster than PyTorch, but not as fast as Lantern.

It is especially interesting to include another framework, DyNet, in this evaluation. DyNet is very similar to PyTorch, being a define-by-run framework, but offers autobatching (Neubig et al., 2017), which dynamically batches similar computation nodes at runtime. We observed that DyNet without autobatching is somewhat slower than Lantern in runtime (labeled DyNetNB), but DyNet with autobatching has approximately a 40% speedup (labeled DyNetB) and ran approximately 20% faster than Lantern. However, we still used batch size 1, so that only autobatching within each training datum is enabled. Our tests show that larger batch sizes actually hurt performance, indicating that DyNet's autobatching heuristics may be improved, and that it is worthwhile to explore autobatching options in Lantern as future work.

## 5  Related Work and Concluding Remarks

Several works from the PL community address the problem of differentiation. Karczmarczuk (2001) presented a functional implementation of differentiation using lazy evaluation that can compute infinite towers of derivatives of higher order. Elliott (2009) developed an implementation of higher-dimensional, higher-order forward-mode automated differentiation (AD) by *calculus on manifolds*. However, in practice forward-mode AD has much higher complexity for machine learning models. The seminal work by Siskind and Pearlmutter (2008) formalized forward- and reverse-mode AD in a functional framework. Several practical projects were developed based on their model, including a flexible differentiable functional programming library called DiffSharp (Baydin et al., 2016), and a differentiable library for natural language processing in Python called Thinc/spaCy[6]. Elliott (2018) provided a generalization of AD based on category theory. However, the model as presented does not cover in-graph control flow, thus limiting the range of application.

The ML community has also worked to bridge the gap between define-by-run frameworks that are easy to use and define-then-run frameworks that are efficient to run. Examples include Tangent (van Merrienboer et al., 2018), which provides AD in Python through the use of source-to- source transformations, and Myia (Breuleux and van Merriënboer, 2017), which implements a first- order gradient operator for a subset of Python (using a dedicated functional representation). Another line of work, AutoGraph (Moldovan et al., 2018), directly stages Python functions into an intermediate representation and subsequently dispatches to different define-then-run frameworks as back-ends including TensorFlow and Lantern.

The history of continuations is recounted nicely by Reynolds (1993).

Compared with related work, our contribution stands out by applying two well-understood PL concepts (delimited continuations and multi-stage programming) to reverse-mode AD, and arriving at a concise, expressive, and efficient backpropagation framework. The underlying ideas are agnostic to the choice of programming languages, thus having the potential to benefit the ML community in broad ways and regardless of implementation language.

### Acknowledgments

This work was supported in part by NSF awards 1553471 and 1564207, DOE award DE-SC0018050, and a Google Faculty Research Award.

## Footnotes

[1]https://github.com/stackless-dev/stackless/wiki

[2]https://github.com/feiwang3311/Lantern

[3]All experiments were run using a single CPU on a cluster with Intel Xeon Platinum 8168 CPUs at 2.70GHz and 0.75 TB RAM per node.

[4] https://github.com/ttpro1995/TreeLSTMSentiment

[5]`https://github.com/tensorflow/fold/blob/master/tensorflow_fold/g3doc/sentiment.ipynb`

[6]`https://github.com/explosion/thinc`

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
