[Supplementary Material · ReproducibilityChecklist.pdf]

# The Machine Learning Reproducibility Checklist  (Version 1.0)

For all **algorithms** presented, check if you include:

- ❑ A clear description of the algorithm.

- ❑ An analysis of the complexity (time, space, sample size) of the algorithm.

- ❑ A link to a downloadable source code, including all dependencies.

For any **theoretical claim**, check if you include:

- ❑ A statement of the result.

- ❑ A clear explanation of any assumptions.

- ❑ A complete proof of the claim.

For all **figures** and **tables** that present empirical results, check if you include:

- ❑ A complete description of the data collection process, including sample size.

- ☑ A link to downloadable version of the dataset or simulation environment.

- ❑ An explanation of how samples were allocated for training / validation / testing.

- ❑ An explanation of any data that were excluded.

- ❑ The range of hyper-parameters considered, method to select the best hyper-parameter configuration, and specification of all hyper-parameters used to generate results.

- ❑ The exact number of evaluation runs.

- ☑ A description of how experiments were run.

- ☑ A clear definition of the specific measure or statistics used to report results.

- ❑ Clearly defined error bars.

- ❑ A description of results with central tendency (e.g. mean) & variation (e.g. stddev).

- ☑ A description of the computing infrastructure used.