[Reviews · NeurIPS 2018]

Reviewer 1



The paper descries Lantern, a framework for automatic differentiation in Scala, based on callbacks and continuation passing style. It compares against PyTorch and TensorFlow on several benchmark tasks. There are two main aspects of the paper: Reverse-mode automatic differentiation with continuations, and code generation via multi-stage programming. The submission does not provide code for the proposed framework, which I don't find acceptable for a paper on a software package. It's unclear to me how the first is different from any other implementation of automatic differentiation via operator overloading. This is usually done using a call-back mechanism, and the delimited continuation seem only syntactic sugar on this concept - which doesn't even enter the user code. The paper also mentions that the proposed solution is different from previous solutions in that it requires no auxiliary data structure. However, OO based approaches typically don't need any additional data structures. The authors explicitly say "Indeed, our computation graph is never reified, but instead remains implicit in the function call stack.", which is typical for OO approaches. The multi-stage programming approach seems more interesting, as it seems Scala with LMS can create compute graphs on the fly within Scala. This is much more flexible than other approaches, like TensorFlow, which have to construct the compute-graph explicitly, or Myia which works on the Python AST, but basically has to reimplement parsing of the Python AST, which restricts Myria to a subset of Python. The LMS approach seems more elegant, but I'm not familiar enough with the internals of scala and LMS to judge the contribution of this paper. The benchmarks comparing this implementation to PyTorch and TensorFlow are quite misleading as most graphs are shown with a batchsize of 1. This is an extreme case of little practical relevance in which both PyTorch and TensorFlow are likely to fail. More realistic cases are also discussed briefly in the text, but not shown in graphics. This seems quite dishonest. One sentence reads "For training time, Lantern with batch size 1 outperformed both PyTorch and TensorFold at batch size 1, and was similar to TensorFold at batch size 20." It seems PyTorch with a larger batchsize is omitted because it faster than Lantern. The paper is written well, though the explanation of shift/reset operators seems cryptic to me (possibly because I'm not overly familiar with scala). It's very unclear however what the contributions of the paper are, and the presentation seems to obscure both the novelty of the OO approach and the results of the benchmark.

Reviewer 2



=== Summary The fields of high-performance computing and systems have played a large role in the recent developments in deep learning. More recently, important research from the compilers and programming languages perspective has been contributing to deep learning, addressing the lack of usability and flexibility that many frameworks have traded for efficiency. This work falls squarely in that category. The authors propose a new implementation approach to reverse mode automatic differentiation. The technique relies on operator overloading, but does not rely on a tape (and the accompanying interpreter that walks the tape backwards). Instead, any primal computation takes a callback to the rest of the forward pass which it calls before performing the adjoint computation, making the "tape" implicit in the function call stack. The authors provide a basic implementation of this idea in Scala, along with a cleaner implementation which relies on delimited continuations (using the shift/reset oeprators) in order to construct the necessary callbacks automatically. Lastly, lightweight modular staging (LMS) is used to reduce the overhead of these abstractions (dual numbers, callbacks, etc.) and generate efficient C++ code. Experiments highlight that this approach is competitive with existing frameworks such as PyTorch and TensorFlow. === Review In the abstract the authors highlight that the use of continuation passing style (CPS) is a key ingredient to their approach. They claim that this can be implemented without any "auxiliary data structures". This claim is repeated in lines 36-37. However, frameworks that rely on having a single global scope (e.g. TensorFlow) or that use closures (e.g. Myia) don't need these data structures either. The same confusion appears in section 2: The text suggests that TensorFlow uses graph as in figure 1, where nodes hold values for both the forward and backward pass. This is not the case. TensorFlow augments the forward graph with additional primitives i.e. it creates a larger graph in which there is no distinction between the forward and backward pass anymore. Hence, the graph itself is never used as an auxiliary data structure that stores intermediate variables and there is no "graph iterator which performs a forward pass first, then a backward pass" (line 75). The remainder of the section does a good job at conveying the main ideas of the paper. In the last two paragraphs the authors compare their approach to Siskind's approach. Although it is true that that transformation is more complex, it must be noted that it operates on lambda calculus, whereas the proposed implementation here relies on (1) compiler support for control delimiters and (2) side effects (i.e. instead of requiring a backpropagator to return partial derivatives to a particular scope it can call tˆ.d+=y.d). The different restrictions under which the transformations apply makes it misleading to compare them without qualifying this more clearly in the text. In section 3 lightweight modular staging (LMS) is introduced as an approach to achieving some of the performance gains that libraries such as TensorFlow can achieve by optimizing and compiling the computation graph. I think the text would possible benefit from a small code sample that involves a loop or conditional here, but overall the text is clear. In the last two sections the authors argue that their approach is applicable to other languages. "Fundamentally" almost anything can be done in any language, but I would have appreciated more insight as to what requirements their approach puts on a language. For example, I think it is far less feasible to implement delimited continuations and LMS in Python, given its dynamically typed, interpreted nature. The authors argue that their approach benefits more from general compiler transformations, but I believe the same is true for e.g. TensorFlow when using the XLA backend, since it lowers the entire graph to LLVM in that case. In section 4 a few simple models are implemented using the proposed approach and benchmarked against PyTorch, NumPy and TensorFlow. I do not think it is necessary to show training curves; that they decrease actually says very little (neural networks and SGD are very robust, and the loss will decrease just as well even if there are mistakes in the gradients). The correctness of gradients can just be checked using finite differences by the authors and be taken for granted by the reader. The runtime benchmarks look promising, but as the authors hint at in the text: The results mainly highlight that if you consider very small models, batch sizes, and datasets, your performance will no longer depend on e.g. your compute kernels, high-level linear algebra optimizations, or your ability to exploit large amounts of parallelism (the things that TensorFlow and PyTorch are optimized for). Instead, your performance will be entirely determined by the overhead of your language and AD implementation, in which case a statically typed language like Scala which will use LMS to compile custom compute kernels will always beat a dynamically typed, interpreted language like Python which will be calling e.g. MKL kernels--the authors failed to mention which BLAS library they linked to--which are optimized for larger workloads on more cores. That said, there is no perfect way to do these benchmarks and for people who care about small workloads in low-resource environments, these results are encouraging. That said, statements such as "Lantern performs comptetitively on cutting edge deep learning applications which push the boundaries of existing frameworks [in] performance" could be qualified, considering that neither the models nor hardware in this benchmark are representative of the common deep learning workloads that existing frameworks are optimized for. In the related work section I think it would be worthwhile to discuss Siskind and Pearlmutter's more recent work on checkpointing, considering that uses both CPS and control operators in the context of reverse mode AD. Finally, some higher-level questions that this work raised for me: The listing in figure 4 suggests that the partial derivative is explicitly initialized to zero during the forward pass. For multidimensional arrays this can lead to the allocation of large amounts of zeros, significantly increasing the peak memory usages. Most existing libraries try to be more clever about this (e.g. Tangent uses a special zero object). Have the authors given this any thought? Higer-order differentiation. An important goal in AD is usually that the gradient operator is closed under its own operation. Also in ML, this is the case e.g. TensorFlow, PyTorch, and more research-oriented libraries such as Tangent, Myia, and Autograd, all have support for higher-order derivatives, and often have support for mixing forward and reverse mode. It is a shame that this is not dicsused at all in this paper. Can a single shift operator be matched to multiple reset operators? === Minor comments 223: Parentheses around citations. 267: lead -> led === Conclusion There are a few misunderstandings in the paper about the internals of TensorFlow that should be addressed, but they don't fundamentally change the authors claim that their approach can naturally support control flow while still supporting multi-stage programming. Other things I would like to see addressed are higher-order gradients, and the supposed applicability of this approach to e.g. Python (which I find hard to believe). Overall I believe the paper is well-written though, the research question relevant, the approach sufficiently novel, and the experimental results, although preliminary, show promise. Hence, I would recommend this work for publication with minor edits.

Reviewer 3



This work describes a novel approach to automatic differentiation as a purely local program transformation on an IR with delimited continuations, and a code-generating deep learning framework built on this idea in Scala that achieves the staged-programming performance benefits of TensorFlow and the natural, language-native expressivity of PyTorch. This represents a compelling convergence of ideas from the programming languages and machine learning communities, and the prototype presented (along with related papers/prototypes by Conal Elliot, Richard Wei, and Olivier Breuleux) suggests numerous promising directions for future work in deep learning frameworks and systems. The paper itself is impressively clear, with my only clarity quibble being that the notation in Figure 3 should be explained more completely for readers not familiar with the PL literature. Some things I noticed/questions I had: - The use of continuations for AD in deep learning, while new to the literature, is not new to the practice. The popular Cython-based natural language processing toolkit SpaCy uses a simple continuation-based AD approach in its thinc (https://github.com/explosion/thinc) deep learning library. This should probably be mentioned. - It's not so much the case that delimited continuations allows AD without any auxiliary data structures like tapes, as that it allows reusing the language-native call stack itself as the Wengert list/tape. - If the only problem with the Pearlmutter and Siskind (2008) approach is that it requires nonlocal program transformation, isn't that solved more simply by extending the single-assignment condition to a single-assigment single-use condition by making computation graph "splits" explicit, which I believe allows the AD to be purely local? - The speed comparison should probably have included DyNet at least for the TreeLSTM, as that's widely seen as the fastest define-by-run framework for NLP tasks.